# Vimentin Promotes the Aggressiveness of Triple Negative Breast Cancer Cells Surviving Chemotherapeutic Treatment

**DOI:** 10.3390/cells10061504

**Published:** 2021-06-15

**Authors:** Marie Winter, Samuel Meignan, Pamela Völkel, Pierre-Olivier Angrand, Valérie Chopin, Nadège Bidan, Robert-Alain Toillon, Eric Adriaenssens, Chann Lagadec, Xuefen Le Bourhis

**Affiliations:** 1UMR9020—U1277-CANTHER—Cancer Heterogeneity, Plasticity and Resistance to Therapies, University of Lille, CNRS, Inserm, CHU Lille, F-59000 Lille, France; marie.winter.etu@univ-lille.fr (M.W.); s-meignan@o-lambret.fr (S.M.); pamela.voelkel@univ-lille.fr (P.V.); pierre-olivier.angrand@univ-lille.fr (P.-O.A.); valerie.chopin@univ-lille.fr (V.C.); nadegebidan@hotmail.fr (N.B.); robert-alain.toillon@univ-lille.fr (R.-A.T.); eric.adriaenssens@univ-lille.fr (E.A.); chann.lagadec@inserm.fr (C.L.); 2Tumorigenesis and Resistance to Treatment Unit, Centre Oscar Lambret, F-59000 Lille, France; 3UFR Sciences, University of Picardie Jules Verne, 33 rue Saint Leu, F-80000 Amiens, France

**Keywords:** triple negative breast cancer, chemotherapy resistance, invasion, cancer stem cells, vimentin, zebrafish xenografts

## Abstract

Tremendous data have been accumulated in the effort to understand chemoresistance of triple negative breast cancer (TNBC). However, modifications in cancer cells surviving combined and sequential treatment still remain poorly described. In order to mimic clinical neoadjuvant treatment, we first treated MDA-MB-231 and SUM159-PT TNBC cell lines with epirubicin and cyclophosphamide for 2 days, and then with paclitaxel for another 2 days. After 4 days of recovery, persistent cells surviving the treatment were characterized at both cellular and molecular level. Persistent cells exhibited increased growth and were more invasive in vitro and in zebrafish model. Persistent cells were enriched for vimentin^high^ sub-population, *vimentin* knockdown using siRNA approach decreased the invasive and sphere forming capacities as well as Akt phosphorylation in persistent cells, indicating that vimentin is involved in chemotherapeutic treatment-induced enhancement of TNBC aggressiveness. Interestingly, ectopic vimentin overexpression in native cells increased cell invasion and sphere formation as well as Akt phosphorylation. Furthermore, vimentin overexpression alone rendered the native cells resistant to the drugs, while *vimentin* knockdown rendered them more sensitive to the drugs. Together, our data suggest that vimentin could be considered as a new targetable player in the ever-elusive status of drug resistance and recurrence of TNBC.

## 1. Introduction

Triple negative breast cancer (TNBC) is a subgroup of breast cancer that is immunohistochemically characterized by the lack of estrogen receptor, progesterone receptor and HER2 (also defined by the lack of HER2 amplification [1]. TNBC represents about 15–20% of annually diagnosed breast cancers and are commonly found in younger women [2]. Even though TNBC is diagnosed by immunohistochemistry, it represents a heterogeneous group of breast cancers and shows strong overlap with the basal-like molecular subtype of breast cancer, i.e., about 70–80% of TNBC cases are basal-like and about 70% of basal-like cancers are triple negative. Due to the lack of specific molecular targets, anthracyclines and taxanes are the chemotherapy of choice for the great majority of TNBC. Most often, patients are first treated with epirubicin (anthracycline) and cyclophosphamide (alkylating agent) and then by paclitaxel (taxane) [3]. These combined and sequential treatments allow for 25–30% of patients to achieve a pathologic complete response (pCR). Unfortunately, about 50% of patients having residual disease will rapidly relapse with frequent metastases to the brain and lung, and only 68% of these patients present a 3-year overall survival [1,4]. Resistance to chemotherapy and rapid recurrence make TNBC a worldwide threat, and considerable effort has been made to elucidate the mechanisms of chemoresistance with the ultimate goal of identifying specific therapeutic targets. It is considered that chemoresistance implies dynamic interactions between tumour cells and their microenvironment, favoring the selection and amplification of preexistent resistant cells as well as the emergence of new resistant cells exhibiting epithelial to mesenchymal transition (EMT) and/or cancer stem cell (CSC) traits [5,6,7]. Numerous signalling pathways have been described to be important drivers of chemoresistance, such as developmental pathways (Transforming Growth Factor-β (TGF-β), Notch, Wnt/β-catenin, hedgehog) and common cancer-related intrinsic pathways (Nuclear Factor-kappa B (NF-kB), Phosphoinositide 3-kinase (PI3K)-Protein Kinase B (AKT)-mechanistic Target Of Rapamycin (mTOR), Janus Kinase (JAK)/Signal Transducer and Activators of Transcription (STAT), Receptor Tyrosine Kinases (RTK)) [8]. Inhibition of these different pathways is the subject of ongoing clinical trials [9,10]. However, the majority of fundamental studies have been conducted with only one drug contrary to the clinical practice. Moreover, most experiments were performed immediately after short-term drug treatment or after chronic treatment with drug concentrations too much higher than that used in clinics [11,12]. Contradictory data have been generated due to experimental designs. For example, paclitaxel is reported to decrease or increase cell migration and invasion of TNBC cells according to the dose and the time of treatment [13,14,15]. In addition, combined treatment with different drugs can result in different results when compared to single drug treatment. Hence, invasion of breast cancer cells has been reported to be enhanced by doxorubicin (anthracycline) but decreased by paclitaxel; however, the invasive ability is no longer modified after sequential treatment with doxorubicin and paclitaxel [16].

In this study, we first treated TNBC cells with epirubicin and cyclophosphamide, and then with paclitaxel to mimic clinical neoadjuvant treatment protocol. We found that persistent TNBC cells surviving the combined and sequential treatment display more proliferative, invasive and stem-like properties. We also demonstrated the importance of vimentin in enhancing TNBC aggressiveness and resistance to combing treatment with chemotherapeutic agents. Our data suggest that vimentin could be considered as a new targetable player in the ever-elusive status of drug resistance and recurrence of TNBC.

## 2. Materials and Methods

### 2.1. Cell Culture

The MDA-MB-231 breast cancer cell line (from ATCC) was maintained in MEM (Minimal Essential Medium, Life technologies, Villebon-sur-Yvette, France) supplemented with 10% inactive FCS (Foetal Calf Serum, Hyclone US; Sigma, Saint-Quentin-Fallonier, France), 1% non-essential amino acid, 40 UI/mL penicillin, 40 µg/mL streptomycin (Biovalley, Nanterre, France) at 37 °C in 5% CO_2_-humidified atmosphere. The SUM159-PT breast cancer cell line (from Asterand Bioscience, Detroit, MI, USA) was grown in Ham’s F12 culture medium supplemented with 5% FCS, 10 mM HEPES, 0.1% insulin, 1 mg/mL hydrocortisone, 40 UI/mL penicillin, 40 µg/mL streptomycin at 37 °C in 5% CO_2_-humidified atmosphere.

### 2.2. Combined and Sequential Treatment of Cells with Epirubicin, Cyclophosphamide and Paclitaxel

In order to mimic clinical neoadjuvant treatment protocol in vitro, cells were first treated with epirubicin (Selleckem, Souffelweyersheim, France) and cyclophosphamide (Selleckem, Souffelweyersheim, France) with a ratio of 1:5 for 48 h, and then treated with paclitaxel (Selleckem, Souffelweyersheim, France) for another 48 h in media containing 1% FCS. Drug concentrations were first screened in a combined and sequential manner, and the concentrations which gave 50% growth inhibition after 96 h of treatment were chosen for experiments. Hence, MDA-MB-231 cells were first treated with 8 nM epirubicin and 40 nM cyclophosphamide for 48 h, and then with 1 nM paclitaxel for another 48 h. SUM159-PT cells were first treated with 40 nM epirubicin and 200 nM cyclophosphamide for 48 h, and then with 2 nM paclitaxel for another 48 h. After the treatment named as ECP (Epirubicin, Cyclophosphamide, Paclitaxel), cells were cultured during 96 h to mimic post-therapy scenario unless otherwise stated. Persistent cells were then harvested for further studies.

### 2.3. Cell Growth Evaluation

#### 2.3.1. Cell Growth in 2D Culture

Cell growth was determined with a hemocytometer four days after ECP treatment.

#### 2.3.2. Clonogenic Assay in 2D Culture

Four days after ECP treatment, 1000 cells were seeded in 6-well plates (Falcon) in culture media containing 10% FCS (for MDA-MB-231 cells) or 5% FCS (for SUM159-PT cells). After 7 days of culture, colonies were stained with crystal violet before counting as previously described [17].

#### 2.3.3. Cell Growth in 3D Culture

Four days after ECP treatment, cells were harvested and 100 cells were resuspended in 50 µL of MEM or F12 media containing 1% FCS. Cell suspension was then mixed 50 µL of growth factor reduced Matrigel (Corning, Borre, France) to form a droplet. The droplet was deposited in a well of 6-well plate (Falcon; Fisher Scientific, Illkirch-graffenstaden, France) and incubated for 30 min in incubator 37 °C, 5% CO_2_-humidified atmosphere. The droplet was then incubated in culture medium containing 10% FCS for 10 days, with culture medium replaced 5 days later. At the end of culture, cells were dissociated from Matrigel with 1 mL of dispase 1 U (Stem cells technologies) and centrifugated (300× *g*, 5 min) to remove the Matrigel. Cells were resuspended in culture medium and counted with a hemocytometer.

### 2.4. Invasion Assay

Invasion assay was performed in Boyden microchambers (Corning) with 8 µm pore size membranes. Four days after ECP treatment, persistent or control cells are trypsinized and resuspended in MEM or F-12 media supplemented with 1% FCS and seeded at a density of 50,000 cells per well precoated with Matrigel at 3 mg/mL. The bottom of chamber contains MEM-10% FCS or F12-5% FCS. After 24 h of culture, invasive cells were fixed and stained as previously described [18]. Pictures were taken under a fluorescent microscope (Nikon, Champigny-sur-Marne, France, 20× and invasive cells were counted with ImageJ software (version 1.51, NIH, Bettesda, MD, USA).

### 2.5. In Vivo Invasive Assay Using Zebrafish Xenograft Model

Zebrafish were handled according to the European Union guidelines for the handling of laboratory animals (Directive 2010/63/EU of the European Parliament and of the Council of 22 september 2010 on the protection of animals used for scientific purpose). The experimental procedures carried out on zebrafish were reviewed and approved by the French Ministry of Higher Education and Research by the Ethics Committee from the Région Hauts-de-France (approval number 2018011725229804v3). Tumour cell injection into transgenic zebrafish line Tg (fli1:GFP) was conducted as previously described [19]. Briefly, control and persistent MDA-MB-231 cells were labelled with 2.5 mg/mL of Vybrant DiI (V-22889, Invitrogen, Villebon-sur-Yvette, France) or Vybrant DiD (V-22887, Invitrogen) according to the manufacturer’s instructions. An equal number of labelled control and persistent cells were mixed and resuspended for a total of 2 × 10^6^ cells in 100 µL of PBS supplemented with 5 mM EDTA before injection in the yolk sac of zebrafish embryos. At 3-day post-injection, zebrafish are fixed with 4% paraformaldehyde for confocal imaging. Fluorescent images were captured with LSM 880 microscope (ZEISS, Oberkoden, Germany). Z-stack slice was made every 5 µm. To avoid any bleed-through of fluorescence signal in multi-dye experiment, imaging was performed using the line-by-line multitrack mode of the confocal scanner. The number of cells invaded in zebrafish embryo tail was counted using ImageJ software. To rule out any Vybrant DiI- or Vybrant DiD-induced artifacts, the invasive essays were repeated with swapped colours.

### 2.6. Sphere Forming Assay

Four days after ECP treatment, cells were trypsinized and cultured in DMEM/F12 (Life technologies) containing 0.4% of Bovine Serum Albumin (BSA, Sigma, Saint-Quentin-Fallonier, France). For MDA-MB-231 cells, AggreWell™400 6-well plates (Stem cells technologies) were used following the manufacturer’s instruction. Briefly, the wells were treated with 2 mL/well of anti-adherence rinsing solution (Stem cells technologies, Cambridge, United Kingdom), and 7000 cells per well were seeded in 5 mL of sphere culture medium. For SUM159-PT cells, low-adherence 96-well plates (Greiner, Les Ulis, France) were used as previously described [20]. After 7 days of culture, sphere number was counted under contrast phase microscope.

### 2.7. Flow Cytometry Analysis

For the detection of cell surface markers clusters of differentiation 49f (CD49f) and clusters of differentiation 24 (CD24), 10^5^ Cells were harvested and resuspended in 100 µL PBS supplemented with 1% FCS. Cells were incubated with FITC fluorescent-conjugated antibodies against CD49f or isotype control (dilution 1:100, BD Pharmingen, Le pont de Claix, France) or PE fluorescent-conjugated antibodies against CD24 or isotype control (dilution 1:100, BD Pharmingen) for 30 min at 4 °C. Cells were then centrifugated at 100× *g* during 5 min and resuspended in PBS for cytometer analysis. For the detection of vimentin, NANOG, OCT4 and SOX2, cells were fixed with 4% Paraformaldehyde (10 min, 20 °C) and permeabilized with 0.1% Triton X100 (10 min, 20 °C) before incubation with PBS 5% BSA (30 min, 20 °C). Staining of cells with vimentin (Abcam, Cambridge, MA, USA), NANOG, OCT4, SOX2 (Biolegend, London, UK) or associated isotypic fluorescent labelled antibodies was performed in PBS-5% BSA for 30 min at room temperature. Cells were then centrifugated and resuspended in PBS and kept on ice before flow cytometry analysis.

For all the experiments, data were collected for 10,000 single cell events. The LSR-Fortessa flow cytometer (BD bioscience) and FlowJo software (VerX.0.7, BD Pharmingen, Le pont de Claix, France) were used for flow cytometry analysis.

### 2.8. RNA Extraction and Real Time-PCR (qRT-PCR)

Total RNA was extracted using a RNeasy Mini Kit (Qiagen, Les Ulis, France) following the manufacturer’s protocol. One µg RNA was reverse transcribed using oligo(dT) with Superscript First Strand Synthesis System (Invitrogen) according to the manufacturer’s recommendations. qRT-PCR was performed with 2 µL of reversely transcribed RNA in a total volume of 10 µL using Sybr Green (Qiagen). The list of used primers (Eurogenetec, Angers, France) is shown in Table 1 and RPLP0 is used as control.

### 2.9. Western Blot Analysis

Western blotting experiment was carried out as previously reported [21]. Membranes were revealed by SuperSignal West Femto Substrate (Thermo Scientific, Illkirch-graffenstaden, France). Chemiluminescence was detected with Fuji LAS-4000 luminescent image analyzer. The antibodies against vimentin (#5741), pan-Akt (#4691), phospho-Akt (Ser-473) (#9271) (from Cell Signalling Technology, Lieden, Netherland) were used at 1:1000 dilution. The anti-actin antibody (from Sigma) was used at 1:10,000 dilution.

### 2.10. Vimentin Knock down and Overexpression

#### 2.10.1. siRNA Transfection

MDA-MB-231 persistent cells (50,000) were seeded in 6-well plates and then transfected with 100 nM siRNA using INTERFERin™ transfection reagent following the manufacturer’s instructions (Polyplus Transfection^®^, Illkirch-graffenstaden, France). Two sequences of siRNA against vimentin (siVIM5, S100302197; siVIM13, S104201890) as well as control (siCTL, S103650325) were from Qiagen. Experiments were performed 72 h after siRNA transfection.

#### 2.10.2. Transient Vimentin Overexpression

Native MDA-MB-231 cells (100,000) were seeded in 6-well plates. After 24 h, cells were transfected with 1 µg/mL of plasmids purchased from Addgene: the pVimentin-PSmOrange-N1 (#31922) contains *vimentin*; the pPSmOrange-N1 (#31898) was used as control. Transfection was performed by using jetOPTIMUS transfection reagent following the manufacturer’s instructions (Polyplus transfection). Experiments were performed 24 h after transfection, and vimentin overexpression was validated by using fluorescent microscopy with pPSmOrange fluorescence (Nikon, 10×) as previously described [22].

### 2.11. Statistical Analysis

Data are presented as the mean values ± standard deviation, taken over at least 3 independent experiments with at least 3 replicates per experiment, unless otherwise stated. Statistical significance was measured with GraphPad Prism software (GraphPad Software ver. 8.0.1.244, Inc., La Jolla, CA, USA) using the paired or unpaired two-tailed student’s *t*-test, with a *p*-value of 0.05 or less considered as statistically significant.

## 3. Results

### 3.1. TNBC Cells Surviving Chemotherapeutic Drug Treatment Exhibit Enhanced Agressiveness in 3D Culture

In order to mimic clinical neoadjuvant treatment in vitro, cells were first treated with epirubicin and cyclophosphamide with a ratio of 1:5 for 48 h, then treated with paclitaxel for another 48 h. After drug withdrawal, cells were maintained in culture for 96 h to mimic post-therapy situation. In these conditions, both MDA-MB-231 and SUM159-PT persistent cells (ECP) exhibited reduced growth when compared to control cells (CTL) (Figure 1A). Similarly, when clonogenic assays were performed to evaluate the capacity of persistent cells to proliferate at low density, we found that MDA-MB-231 and SUM159-PT persistent cells formed less colonies than control cells (Figure 1B).

We next determined cell behaviours using Matrigel 3D culture. As shown in Figure 1C, after 10 days of culture, cell number was increased to about 40–50% of in both MDA-MB-231 and SUM159-PT persistent cells (Figure 1C). Interestingly, persistent cells tended to disperse more in Matrigel with the formation of more invasive fronts (Figure 1D), suggesting an increased invasive capacity of these cells. In order to confirm these observations, we quantified invasive cells using Boyden microchambers precoated with Matrigel (Figure 1E). The capacity of invasion was increased in MDA-MB-231 and SUM159-PT persistent cells compared to control cells.

Cancer stem cells (CSCs) are known to be involved in both therapy resistance and metastasis [23,24]. To investigate whether persistent cells were enriched in CSCs, we performed the golden standard in vitro functional assay, i.e., sphere forming capacity, to analyze the ability of cancer stem cells to propagate under anchorage-independent condition in defined medium (Figure 1F). An increase of sphere forming units was observed in MDA-MB-231 and SUM159-PT persistent cells compared to control cells, suggesting an enrichment of CSCs in cells surviving ECP treatment.

### 3.2. Persistent Cells Display Enhanced Invasion in Zebrafish

The observed in vitro behaviours of persistent cells prompted us to evaluate their metastatic potential in vivo by using zebrafish model. For this, MDA-MB-231 control and persistent cells were micro-injected in yolk sac of 2 days post-fertilization zebrafish embryos, invasive cells in zebrafish embryo tail were then determined at 3 days post-injection. As shown in Figure 2, more invasive cells were found in MDA-MB-231 persistent cells when compared to control cells, indicating that persistent MDA-MB-231 cells were more invasive in vivo.

### 3.3. Persistent Cells Present Enhanced Expression of Cancer Stem Cell Markers and Vimentin

Increased sphere forming capacity of persistent TNBC cells prompted us to investigate the expression of pluripotency factors NANOG, OCT4, SOX2. As shown in Figure 3A, at mRNA level, *NANOG* was increased in both MDA-MB-231 and SUM159-PT persistent cells, *OCT4* was not modified either in MDA-MB-231 or in SUM159-PT persistent cells, *SOX2* was decreased in SUM159-PT persistent cells. At protein level, flow cytometry analysis revealed that all MDA-MB-231 persistent cells expressed high levels of NANOG (NANOG^high^) (Figure 3B), while no difference was observed in SUM159-PT cells (data not shown). No modification of OCT4 and SOX2 was observed either in MDA-MB-231 or in SUM159-PT persistent cells when compared to control cells (data not shown). Apart from these pluripotency factors, membrane proteins CD24 and CD49f have also been described for their roles as cancer stem cell markers and/or in tumour resistance to chemotherapy [25,26]. We found that the levels of CD24 were not modified either in MDA-MB-231 or in SUM159-PT persistent cells (data not shown). By contrast, the CD49f-positive subpopulation was increased in MDA-MB-231 persistent cells (Figure 3C) but not in SUM159-PT persistent cells (data not shown).

Finally, vimentin, a marker of epithelial to mesenchymal transition, is also increasingly described to be involved in metastasis process [27,28]. Increased vimentin expression has been reported in a low adherent and more metastatic sub population of MDA-MB-231 cancer cells [29]. We then investigated the expression of vimentin in cells surviving ECP treatment. qRT-PCR and Western blot analyses showed that vimentin was increased in persistent cells (Figure 4A,B). Interestingly, while the proportions of vimentin-positive cells were similar in persistent vs. control cells (Figure 4C,D), persistent cells exhibited a significant increase of vimentin^high^ sub-population in both MDA-MB-231 and SUM159-PT cell lines.

### 3.4. Vimentin Down-Regulation Reduces Invasion and Sphere Formation as Well as Akt Phosphorylation in MDA-MB-231 Persistent Cells

To investigate whether vimentin regulates invasive and/or stemness properties in persistent cells, vimentin was silenced with two sequences of siRNA. Vimentin knockdown (Figure 5A,B) resulted in reduced cell invasion (Figure 5C) and sphere formation (Figure 5D), indicating the involvement of vimentin in the increased aggressiveness of persistent cells.

Akt pathway is implicated in diverse processes including the regulation of stemness and invasive properties of tumour cells [30]. Moreover, it has been reported that AKT activation increases soft-tissue sarcomas cell motility and invasiveness at least partially through its interaction with vimentin [31]. We investigated if vimentin inhibition could impact Akt activation. We found that persistent cells exhibited an increased level of phosphoAkt at serine 473 (pS473Akt) compared to control cells (Figure 5E). Vimentin knockdown in persistent cells decreased pS473Akt (Figure 5F), suggesting that vimentin could be involved in Akt activation of MDA-MB-231 persistent cells.

### 3.5. Vimentin Promotes Aggressive Phenotype and Drug Resistance of Native MDA-MB-231 Cells

To further verify the involvement of vimentin in the promotion of breast cancer cell aggressiveness, native MDA-MB-231 cells were first transfected with pVimentin-PSmOrange-N1 to transiently overexpress vimentin. As shown in Figure 6, vimentin overexpression (Figure 6A) resulted in increased cell invasion and sphere formation (Figure 6B,C). In addition, both Akt and pS473Akt were enhanced in vimentin overexpressing cells (Figure 6D). Of note, vimentin overexpression rendered cells resistant to chemotherapeutic agents (Figure 6E), while vimentin knockdown rendered cells more sensitive to the drugs (Figure 6F). These results indicate that vimentin plays a crucial role in promoting breast cancer cell aggressiveness and resistance to chemotherapeutic agents.

## 4. Discussion

In contrast with high chemotherapy response, triple negative breast cancer (TNBC) has become a worldwide threat because of its rapid metastatic relapse after treatment. Molecular and cellular mechanisms underlying TNBC resistance to single treatment with anthracyclines or taxanes have been largely documented in vitro and in vivo [23,24,32,33,34]. However, the consequence of combined and sequential treatment mimicking clinical protocol still remains poorly described. In this study, we first co-treated TNBC cells with epirubicin and cyclophosphamide, and then with paclitaxel to mimic clinical neoadjuvant treatment in order to better understand chemotherapy-induced cell plasticity, potentially responsible for disease recurrence.

We found that even though persistent cells exhibited less proliferative ability in 2D culture, these cells became more proliferative under Matrigel 3D culture conditions, and they were also more invasive both in vitro and in vivo in zebrafish embryos. In addition, persistent cells seemed to be enriched for cancer stem cells (CSCs) or their progenitors, as revealed by increased tumour sphere formation and increased expression of CSC markers including NANOG and CD49f. Therefore, it is possible that in the clinical scenario, CSCs can be generated in the host tumour microenvironment as early as the end of first round of chemotherapy treatment. Our findings are in accordance with several previous works describing the enrichment of CSCs following treatment with one chemotherapeutical agent [5,35,36]. In particular, it has been reported that NANOG expression is increased in MDA-MB-231 cells treated with paclitaxel or doxorubicin [35] or in MDA-MB-231 cells made resistant to paclitaxel after exposition to high dose [36]. NANOG expression is associated with a poor OS and DFS in breast cancer patients [37]. NANOG induction in association with WNT1 pathway induces tumorigenesis in murine mammary gland and enhances mammary tumour metastasis [38]. Thus, apart from its well-established role as a pluripotency transcriptional factor, our findings together with the published data suggest that NANOG may be a marker of therapy resistance and cancer progression.

CD49f is reported to be a robust CSC marker in several cancers including breast cancer [39]. The presence of CD49f-positive cells in breast cancer is associated with a poor clinical outcome [40,41]. Recently, using breast cancer patient-derived orthoxenografts (PDXs), Gomez-miragaya et al. [25] have demonstrated that CD49f-positive sub-population exhibits tumour-initiating ability and increased resistance to taxanes. This sub-population is present in the initially sensitive TNBC and expands during continued exposure to the drug in vivo, contributing to taxane resistance and tumour recurrence. In accordance with these findings, we found that MDA-MB-231 persistent cells surviving short-term sequential exposure of chemotherapeutic agents displayed high frequency of CD49f positive sub-population. Interestingly, it has been reported that down-regulation of CD49f in CSCs isolated from mammary tumours of *Brca1*-mutant mouse model induces reduced cell migration, suggesting a role of CD49f in this process [42].

Another important finding in this study is the increased subpopulation highly expressing vimentin in persistent TNBC cells. Vimentin is an intermediate filament of basal-like and mesenchymal cells and is considered as a canonical marker and an important regulator of epithelial mesenchymal transition (EMT) in several types of cancers including breast cancer [27,43].

EMT and stemness regulation share a lot of common actors and signalling pathways. Peuhu et al. [44] have recently described vimentin as a positive regulator of stemness in mouse mammary gland and breast cancer cells. They observed that in adult *vimentin* knockout mice (*Vim*^−/−^), mammary ductal outgrowth is delayed, with dilated ducts and a reduced ratio of basal-to-luminal mouse mammary epithelial cells. Isolated *Vim*^−/−^ mouse mammary epithelial cells form fewer mammospheres and basal-like organoids in vitro than their wild-type counterparts. Consistent with these findings, we showed here that *vimentin* silencing decreased sphere forming capacities of persistent TNBC cells while vimentin overexpression increased sphere forming capacities in cells not exposed to combined and sequential chemotherapeutic treatment.

High vimentin expression was significantly associated with a reduced survival in TNBC [45,46,47], suggesting that the increased expression of vimentin could lead to high invasiveness of TNBC. Accumulating data highlight the multifaceted role of vimentin in the regulation of actin dynamics [48,49] integrin turnover and as a signalling scaffold that can bind different intracellular proteins [50,51,52]. Indeed, vimentin regulates gene expression and is required for EMT induced by Ras, Slug and TGFβ in breast cancer cells [53,54]. Ectopic expression of oncogenic H-Ras-V12G and Slug induces the expression of vimentin, which in turn upregulates the expression of the membrane tyrosine kinase receptor Axl, leading to enhanced cell migration and invasion in vitro, and lung metastasis in xenografted mouse model [54]. Accordingly, *vimentin* silencing decreases pulmonary metastases in a pre-diabetic mouse model of mammary tumor progression [55]. Virtakoivu et al. [53], have reported that Slug, active ERK and vimentin are co-expressed at the invasive edges of TNBC; the authors have demonstrated that vimentin expression is critical for maintaining high ERK activity in TNBC cells. The reciprocal vimentin-ERK regulation facilitates ERK phosphorylation of Slug, leading to Axl expression and increased cell invasion. Furthermore, *vimentin* silencing in these cells also impairs xenografted tumor growth on chick embryo chorioallantoic membranes, indicating a role of vimentin in cell growth [53]. Here, we showed that the increased expression of vimentin in persistent TNBC cells was associated with Akt activation. *Vimentin* silencing in MDA-MB-231 persistent cells decreased the level of pS473Akt while vimentin overexpression in native MDA-MB-231 cells increased the levels of both Akt and pS473Akt, indicating that vimentin could participates to Akt activation. Indeed, vimentin has been described to interact with MAP2K4 to activate AKT pathway, leading to increased proliferation and invasion of breast cancer cells [56]. On the other hand, it has been shown that AKT activation induces increased vimentin phosphorylation, leading to the protection of vimentin against caspase-induced proteolysis and increased cell survival and migration in soft tissue sarcoma cells [57]. Taken together, these data indicate that vimentin can functionally interact with Akt to regulate cancer cell growth, migration and invasion. In this work, we found that vimentin overexpression itself rendered MDA-MB-231 breast cancer cells resistant to chemotherapeutic agents, while vimentin knockdown rendered cells more sensitive to the drugs, indicating the involvement of vimentin in drug resistance.

Vimentin is not only expressed in cancer cells of different tissue origin, it is also expressed in endothelial cells and has been increasingly shown to be involved in vasculature branching through several mechanisms including the regulation of Notch signalling pathways [58,59]. Moreover, *vimentin* silencing in nasopharyngal cancer cells inhibits the expression of pro-angiogenic factors VEGF and CD31 as well as the formation of pulmonary metastasis in nude mice [60]. Recently, vimentin has been proposed as a biomarker for the use of cancer treatment and a druggable protein [43]. For example, simvastatin, a small-molecules inhibitor of cholesterol biosynthesis pathway, has been shown to induce apoptosis of MDA-MB-231 breast cancer cells through direct interaction between simvastatin and vimentin [61]. Similarly, withaferin A, a natural product, can also interact with vimentin to inhibit both breast cancer cell growth and neovascularization in vivo [62,63,64].

In conclusion, by short-term combined and sequential exposure to epirubicin, cyclophosphamide and paclitaxel to mimic clinical neoadjuvant treatment, we found that persistent TNBC cells display invasive and stem-like properties with increased expression of NANOG, CD49f and vimentin. Vimentin is involved in invasive and sphere forming capacities of persistent TNBC cells. Our findings, together with the published data indicate that vimentin plays an important role in different cellular processes responsible of tumour development and disease relapse. More studies are clearly needed to better understand the underlying mechanisms and to evaluate whether vimentin high sub-population can be manipulated to unveil the ever-elusive status of tumour drug resistance and recurrence. In addition, it would be important to identify common fragile nodes of vimentin, NANOG and CD49f signalling in a more integrated manner in order to evaluate their interest as markers and or new therapeutic targets in triple negative breast cancer.

## Figures and Tables

**Figure 1 cells-10-01504-f001:**
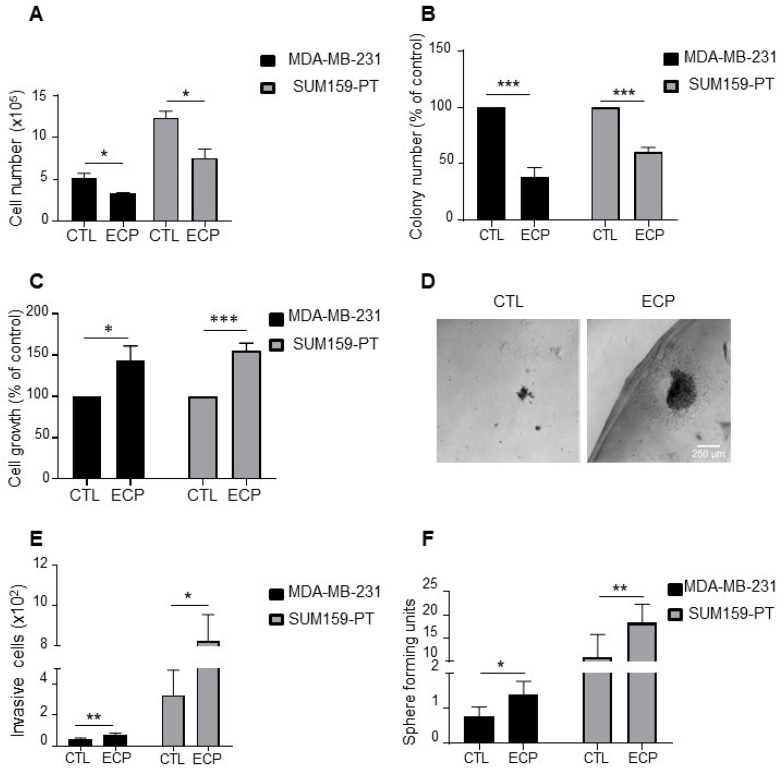
Phenotype characterization of TNBC persistent cells. MDA-MB-231 cells were treated with 8 nM epirubicin and 40 nM cyclophosphamide for 48 h, and then with 1 nM paclitaxel for another 48 h. SUM159-PT cells were treated with 40 nM epirubicin and 200 nM cyclophosphamide for 48 h and then with 2 nM paclitaxel for another 48 h. After drug treatment (ECP), cells were maintained in culture for 4 days to mimic post-therapy scenario. (**A**) Cell count under 2D conditions with a hemocytometer four days after treatment. (**B**) Clonogenic growth. Four days after ECP treatment, cells were harvested and cultured in medium containing 10% FCS for 7 days. Colony formation was evaluated after crystal violet staining. (**C**,**D**) Cell growth in Matrigel. Four days after ECP treatment, cells were seeded in a mix solution of media containing 1% FCS and Matrigel (v:v, 1:1) to form a droplet, and then cultured in media containing 10% FCS for 10 days. At the end of culture, cells were extracted from Matrigel and counted with a hemocytometer. (**C**) Cell growth quantification and (**D**) Illustration of cells cultured in Matrigel. Scale bar is 100 µm. (**E**) Four days after treatment, MDA-MB-231 and SUM159-PT cells were seeded in the top of Boyden microchambers precoated with Matrigel. Invasive cells were counted following 24 h of culture. (**F**) Sphere formation of persistent cells. Sphere forming units, calculated as the number of spheres over the number of initially seeded cells. Quantitative graphics correspond to at least 3 independent experiments and the illustrations are representative of 3 independent experiments. *, *p* <0.05; **, *p <* 0.01; ***, *p* < 0.001. Unpaired Student *t*-test.

**Figure 2 cells-10-01504-f002:**
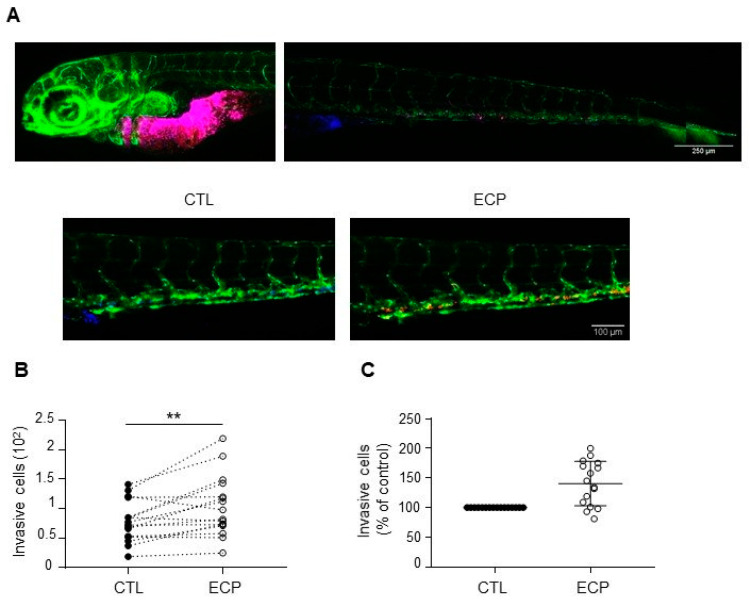
Invasion of MDA-MB-231 cells in xenografted zebrafish embryo. Equal quantities of control and persistent cells were mixed and then co-injected into in the yolk sac of zebrafish embryos before confocal analyses at 3 days post-injection as described in materials and methods. (**A**) Images of embryos at 3 days post-injection. Top panel shows merged images of control (blue) and persistent cells (red) at the injected site and in the tail of an embryo. Scale bar is 500 µm. Bottom panel shows separate images of control (blue) or persistent cells (red) in an embryo tail. Scale bar is 100 µm. Pictures were taken under a Zeiss confocal microscope (10×). (**B**) Graphic showing the number of invasive cells (persistent cells vs. control) in each zebrafish embryo (*n* = 16). (**C**) Relative quantification of invasion. Invasion of persistent dells was expressed as the percent of control cells in each embryo (*n* = 16). Illustrations and quantitative graphics are representative of 2 independent experiments, with 16 zebrafish embryos analyzed in each experiment. **, *p* < 0.01. Paired Student *t*-test.

**Figure 3 cells-10-01504-f003:**
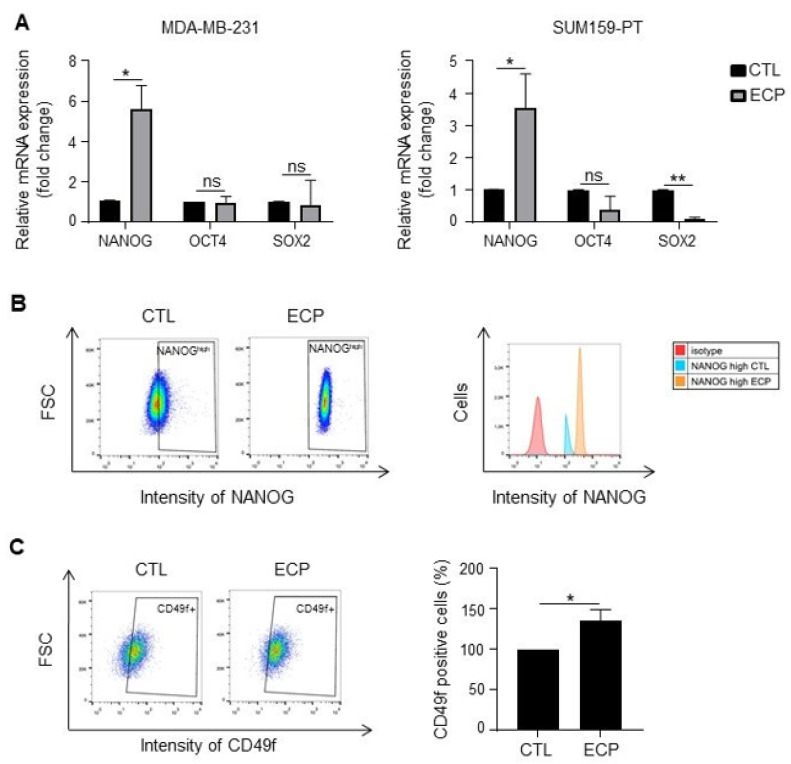
Expression of cancer stem cell markers in TNBC cells. Analyses were performed four days after ECP treatment. (**A**) Real-time PCR analysis of mRNA levels of pluripotency factors *NANOG*, *OCT4* and *SOX2* in MDA-MB-231 and SUM159-PT cells. *RPLP0* mRNA was used to normalize variability in template loading. Data from control cells were set as 1, relative mRNA expression was evaluated by fold-change. (**B**,**C**) Flow cytometry analysis of cancer stem cell markers NANOG and CD49f in MDA-MB-231 cells. Cells were incubated with antibodies against NANOG and CD49f, and then analyzed by flow cytometry with FITC or APC filter, respectively. Isotype antibodies were used as controls. (**B**) Flow cytometry analysis of NANOG^high^ population. Cells with fluorescence intensity ≥ log 10^2^ were considered as NANOG^high^. Data were representative of 3 independent experiences (**C**) Flow cytometry analysis of CD49f positive sub-population. Quantitative graphics correspond to 3 independent experiments and illustrations are representative of 3 independent experiments. ns, not significant; *, *p* < 0.05; **, *p* < 0.01. Unpaired Student *t*-test.

**Figure 4 cells-10-01504-f004:**
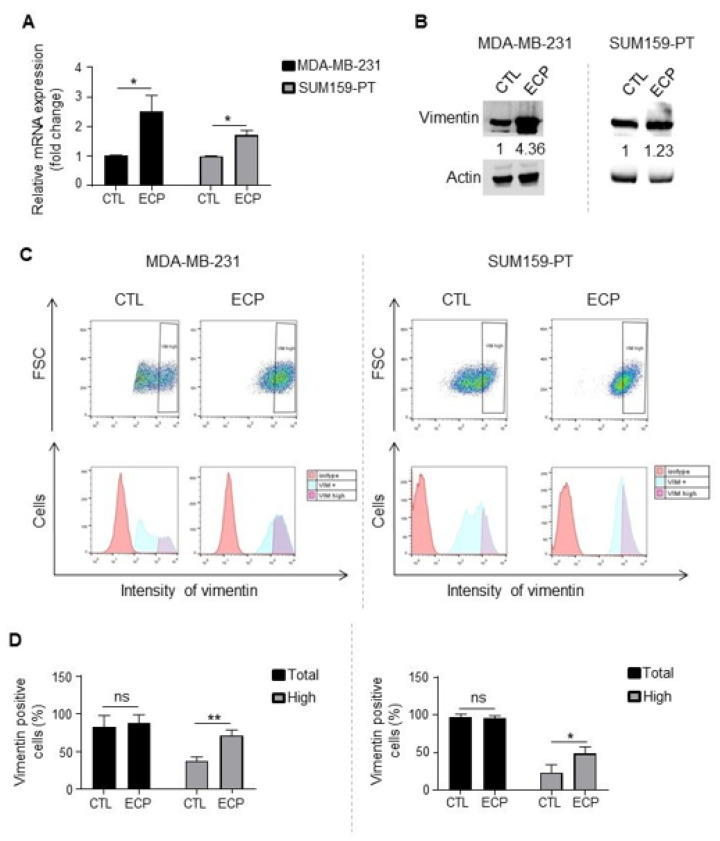
Vimentin expression in TNBC cells. Analyses were performed four days after ECP treatment. (**A**) Real-time PCR analysis of mRNA levels of *vimentin*. *RPLP0* mRNA was used to normalize variability in template loading. Data from control cells were set as 1, relative mRNA expression was evaluated by fold-change. (**B**) Western blot analysis of vimentin. Actin was used as loading control. (**C**) Flow cytometry analysis of vimentin^high^ sub-population. Cells were incubated with an antibody against vimentin and analyzed by flow cytometry with APC filter. Isotype antibody was used as control. Cells with APC fluorescence intensity ≥ log 10^3^ were considered as vimentin^high^. (**D**) Quantification of vimentin positive and vimentin^high^ sub-population in MDA-MB-231 cells (left panel) or SUM159-PT (right panel) cells. Quantitative graphics correspond to 3 independent experiments and illustrations are representative of 3 independent experiments. ns, not significant; *, *p* < 0.05; **, *p* < 0.01. Unpaired Student *t*-test.

**Figure 5 cells-10-01504-f005:**
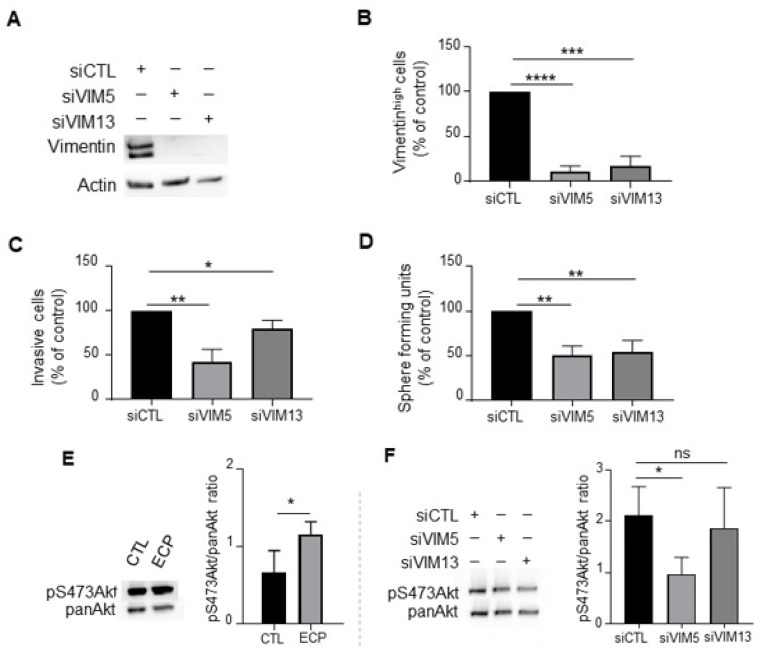
Effects of *vimentin* silencing on invasion, sphere formation and Akt phosphorylation in MDA-MB-231 persistent cells. Four days after ECP treatment, MDA-MB-231 persistent cells were seeded and cultured for 24 h before transfection with 2 sequences of siRNA targeting *vimentin* (VIM5, VIM13). Three days after transfection, vimentin levels were determined by Western blotting (**A**), or flow cytometry analysis to quantify vimentin^high^ sub-population (**B**). Cells with fluorescence intensity ≥ log 10^3^ were considered as vimentin^high^. (**C**) Effect of *vimentin* silencing on invasion of persistent cells. Three days after transfection with siRNA against *vimentin*, MDA-MB-231 persistent cells were seeded in the top of Boyden microchambers precoated with Matrigel. Invasive cells were counted following 24 h of culture. (**D**) Effect of *vimentin* silencing on sphere formation of persistent cells. Three days after transfection with siRNA against *vimentin*, MDA-MB-231 persistent cells were cultured in suspension in defined medium as described in materials and methods. The number of spheres was counted under a contrast phase microscope after 7 days of culture. (**E**) Western blot analysis of pS473Akt in control and persistent cells. Actin was used as loading control. (**F**) Western blot analysis of pan-Akt and pS473Akt in *vimentin* silenced persistent cells. Pan-Akt was used as loading control. Quantitative graphics correspond to 3 independent experiments and illustrations are representative of 3 independent experiments. ns, not significant; *, *p* < 0.05; **, *p* < 0.01; ***, *p* < 0.001; ****, *p* < 0.00001. Unpaired Student *t*-test.

**Figure 6 cells-10-01504-f006:**
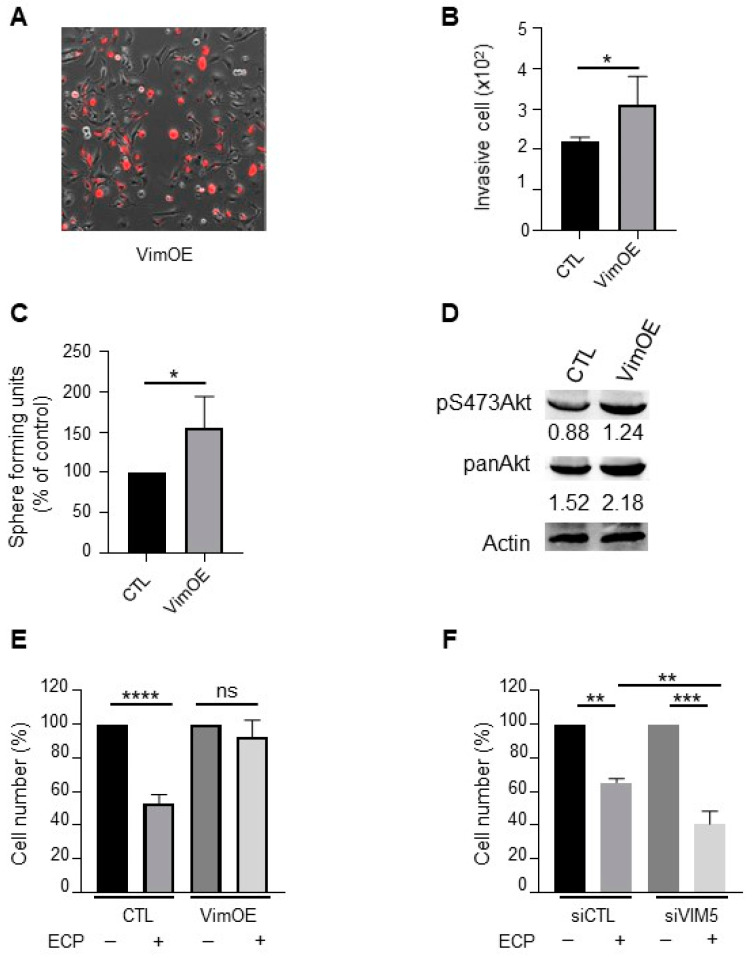
Effect of vimentin overexpression on invasion, sphere formation, Akt activation and drugs sensitivity in MDA-MB-231 wild type cells. MDA-MB-231 wild type cells were seeded and cultured 24 h before transfection with plasmid overexpressing vimentin (VimOE) or control (CTL). The day after transfection, vimentin expression was evaluated by fluorescent microscopy with the texas red filter (**A**; Nikon, 10x). All experiments were performed 24 h after transfection. (**B**) Effect of VimOE on invasion. VimOE cells were seeded on the top of Boyden micro-chambers precoated with Matrigel. Invasive cells were counted following 24 h of culture. (**C**) Effect of VimOE on sphere formation. VimOE cells were cultured in suspension in defined medium described in materials and methods. The number of spheres was counted under a contrast phase microscope after 7 days of culture. (**D**) Western blot analysis of pan-Akt and pS473Akt in MDA-MB-231 VimOE cells. Actine was used as loading control. (**E**,**F**) Vimentin effect in drug sensibility. MDA-MB-231 wild type cells were seeded and cultivate 24 h before transfection with siRNA or plasmid targeted vimentin expression level similar as is previously described. The day after transfection, cells inhibited or overexpressed vimentin were seeded and then treated in first during 48 h with 8 nM epirubicin and 40 nM cyclophosphamide, and then 1 nM paclitaxel for another 48 h in medium containing 1% FCS. After the 96 h of treatment, the cells were count with hemocytometer. (**E**) MDA-MB-231 wild type cells silencing for vimentin (siVIM5, siVIM13) or with the siRNA control (siCTL). (**F**) MDA-MB-231 wild type cells overexpressing vimentin (VimOE) or control (CTL). Quantitative graphics correspond to 3 independent experiments. ns, not significant; *, *p* < 0.05; **, *p* < 0.01; ***, *p* < 0.005; ****, *p* < 0.0001. Unpaired Student *t*-test.

**Table 1 cells-10-01504-t001:** Primer sequences for qPCR.

Gene Name	Forward Sequence	Reverse Sequence
Nanog	5′GTG-ATT-TGT-GGG-CCT-GAA-GA3′	5′ACA-CAG-CTG-GGT-GGA-GA3′
Oct4	5′GAA-GGA-TGT-GGT-CCG-AGT-GT3′	5′GTG-AAG-TGA-GGG-CTC-CCA-TA3′
Sox2	5′AAC-CCC-AAG-ATG-CAC-AAC-TC3′	5′CGG-GGC-CGG-TAT-TTA-TAA-TC3′
Vimentin	5′TCT-AGG-AGG-AGA-TGC-GG3′	5′GGT-CAA-GAC-GTG-CCA-GAG-AC3′
RPLP0	5′GCG-ACC-TGG-AAG-TCC-AAC-TA3′	5′TGT-CTG-CTC-CCA-TGA-AG3′

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
