# Peer review of "Vimentin Promotes the Aggressiveness of Triple Negative Breast Cancer Cells Surviving Chemotherapeutic Treatment"

_cells, 2021, doi:10.3390/cells10061504_

Round 1

Reviewer 1 Report

All my previous comments were addressed by the authors. 

Author Response

Thank you very much for your valuable comments

Reviewer 2 Report

This manuscript by Winter M et al., describes clearly and thoroughly the impact of treatments of human triple-negative breast cancer (TNBC) cells with drugs in a sequence that mimics clinical neoadjuvant treatment protocol. The studies clearly show that the TNBC cells display invasive and stem-like properties with increased expressions of NANOG, CD49f, and Vimentin. Together, the studies strongly indicate that Vimentin plays an important role in a variety of cellular processes responsible for tumor development and disease relapse. The work should capture the interest of the wide TNBC audience and it is timely.  However, a few corrections are required.

  1. Abstract, line 27: change "--- formation well as Akt --" to "--- formation as well as Akt --".
  2. Figure 1G is missing.
  3. Line 275: change "Br contrary," to "By contrast,"
  4. Line 466. Insert none after 5. Patents:

Author Response

Thank you for your valuable remarks. All the suggested corrections in the text are done.  The legend of Figure 1 is now corrected to correspond to the Figure.

Reviewer 3 Report

The manuscript entitled “Vimentin promotes the aggressiveness of triple negative breast cancer cells surviving chemotherapeutic treatment” reports evidence to indicate that vimentin is involved in cancer aggressiveness in triple negative breast cancer (TNBC) cells. In in-vitro treated cells, among other sound evidence, Vimentin knockdown and overexpression using siRNA approach decreased and increased the invasive and sphere forming capacities respectively. The limitations of the evidence were not discussed to support the conclusion in terms of the molecular mechanisms of Vimentin effects on TNBC. In particular, since Vimentin is considered as an intermediate filament know to be expressed in endothelial cells and perhaps could be involved in angiogenesis and vascular invasion in cancer metastasis. Addressing those issues in the discussion will strengthen the manuscript further.  

Author Response

Thank you very much for your valuable suggestion. The involvement of vimentin in the regulation of endothelial cells and cancer angiogenesis /metastasis is now added in the discussion with corresponding references (lines 475-484).